# The Thoracic Inlet Length as a Reference Point to Radiographically Assess Cardiac Enlargement in Dogs with Myxomatous Mitral Valve Disease

**DOI:** 10.3390/ani13162666

**Published:** 2023-08-18

**Authors:** David Marbella Fernández, Verónica García, Alexis José Santana, José Alberto Montoya-Alonso

**Affiliations:** 1Faculty of Veterinary Medicine, University of Las Palmas de Gran Canaria, 35413 Las Palmas de Gran Canaria, Spain; 2CEU Small Animal Hospital, 46115 Valencia, Spain; 3Fénix Hospital Veterinario, 03202 Elche, Spain; 4Anicura Albea, 35011 Las Palmas de Gran Canaria, Spain; 5Internal Medicine, Faculty of Veterinary Medicine, Research Institute of Biomedical and Health Sciences (IUIBS), University of Las Palmas de Gran Canaria, 35413 Las Palmas de Gran Canaria, Spain; alberto.montoya@ulpgc.es

**Keywords:** thoracic inlet, canine, myxomatous mitral valve disease, cardiac enlargement

## Abstract

**Simple Summary:**

The present study investigates the hypothesis that the thoracic inlet heart score (TIHS), a recently described method to assess the cardiac silhouette on dogs’ chest radiographs, can be used to identify dogs with cardiac enlargement secondary to myxomatous mitral valve disease (MMVD). This method uses the thoracic inlet length as a reference point. Degenerative mitral valve disease is the most common acquired cardiac disease in dogs and radiographic studies are important in its diagnosis and follow up. The VHS and VLAS are recommended radiographic measurements to use in the staging of the disease. The TIHS method is simple to perform and provides practitioners with another tool when evaluating dogs with clinical signs compatible with cardiac disease. Comparing the results obtained from clinically healthy dogs and dogs in different MMVD Stage, we found that this method identified dogs with cardiac enlargement secondary to MMVD, and it could differentiate dogs in different MMVD stage.

**Abstract:**

The diagnostic value of the vertebral heart size (VHS) in dogs with mitral valve degeneration (MVD) is compromised when middle thoracic vertebral anomalies are present. The objective of this study was to assess the use of the thoracic inlet heart score (TIHS) to identify left heart enlargement (LHE) secondary to MVD. The cardiac silhouette of 50 clinically healthy dogs and 106 MVD dogs in different stages was assessed on a right lateral chest radiograph. The TIHS and VHS value were calculated for each patient and compared. The TIHS was significantly different between the control dogs and the dogs with MMVD, increasing with disease stage, control 2.91 ± 0.23, Stage B1 2.98 ± 0.36, B2 3.25 ± 0.34, and C 3.53 ± 0.36, *p* < 0.05. A THIS ≥3.3 showed 69% sensitivity and 81% specificity to identify LHE. The TIHS showed moderate correlation with the VHS, LA/Ao, and LVIDDN 0.59, 0.42, and 0.62, respectively. The intraobserver and interobserver agreement were almost perfect, 0.96, and substantial, 0.73. The TIHS method can be used to identify LHE secondary to MMVD on dogs’ thoracic radiographs.

## 1. Introduction

Myxomatous mitral valve disease (MMVD), the most common cardiac disease in dogs [1], is diagnosed based on clinical, radiographic, and echocardiographic findings. MMVD causes left heart chambers enlargement. Echocardiography is the gold standard diagnostic tool for the identification of mitral valve thickening and/or valve prolapse and regurgitation that characterize the disease. It is also used for staging the disease in conjuction with auscultation and radiography [1]. Radiography can also assist the small animal practitioner in the identification of patients with cardiac enlargement when echocardiography is not available. Radiography is also crucial for the diagnosis of congestive heart failure in dogs with symptoms associated with the disease (tachypnea, dyspnea, and/or cough) [2].

Different radiographic methods to assess the dog heart size have been described: intercostal spaces [3], VHS [4], cardiothoracic ratio [5], MHS [6], HSVR [7], TIHS [8], and more specific for the left atrium Tracheal-bifurcation angle [9], VLAS [10], RLAD [11], Br-Spine [12], MVLAS [13], m-VLAS [14]. The diagnostic value of some of these methods, VHS [2,15,16,17,18,19,20], MHS [21], VLAS [10,16,18,19,20,22,23], RLAD [11,16,20,23], MVLAS [13], m-VLAS [14], Br-Spine [12], Tracheal-bifurcation angle [9] to discriminate between dogs suffering mitral valve disease with and without cardiac enlargement has been studied.

The VHS described in 1995 [3] normalizes the sum of the cardiac long and short axes to the length of the midthoracic vertebrae, with a reference value of 9.7 ± 0.5; an upper limit for normal heart size in most breeds of ≤10.5 v was proposed. The same study monitored one dog with mitral regurgitation during a 3.5-year period and observed that the VHS increased with time. The authors concluded that the major uses of the VHS would be to identify cardiomegaly and to monitor its progression over time in a case-basis [3]. That study also considered that different breeds might have different normal upper limits. Since then, many breed-specific VHS reference values have been published [24,25,26,27,28,29,30,31,32,33,34,35,36,37,38,39,40].

ACVIM consensus guidelines for the diagnosis and treatment of MMVD [1] proposed a VHS ≥11.5 or a breed-adjusted VHS or evidence of increasing interval change in radiographic cardiac enlargement patterns in the absence of echocardiography as a predictor of Stage B2 dogs.

However, the VHS is affected by the presence of middle thoracic vertebral anomalies, interobserver differences in reference point selection, and transformation into vertebral units [41]. The thoracic inlet length has been used previously as a reference point to assess tracheal diameter in brachycephalic and non-brachycephalic dogs [42,43,44]. Its use to normalize cardiac size could overcome some of the limitations related to the presence of vertebral malformations and conversion to vertebral units. Thus, a method to assess cardiac size using the thoracic inlet length as a reference point has been described recently, namely the thoracic inlet heart size (TIHS) [8]. For a general population of healthy dogs, its mean value was 2.86 ± 0.27, and a value <3.2 was proposed as an upper limit for clinically healthy dogs. The TIHS was not different between healthy chihuahua and Yorkshire terrier, two dog breeds commonly affected by MMVD. We hypothesized that the TIHS could discriminate between MMVD dogs with cardiac remodeling from dogs without it, and healthy dogs.

The objective of this study was to determine if the TIHS method could discriminate between clinically healthy dogs and dogs with different stages of MMVD. Also, we studied the TIHS variability in dogs of the chihuahua breed diagnosed with MMVD in different stages. Correlation of the TIHS with the VHS, as well as intra- and interobserver agreement, was assessed. We finally suggested a TIHS reference value for dogs with cardiac remodeling.

## 2. Materials and Methods

### 2.1. Animals

The study design was a retrospective observational investigation conducted at Anicura Albea Small Animal Hospital. Therefore, no institutional animal care and use approval were required. Records of client-owned dogs admitted at the hospital from March 2021 to June 2022 were reviewed. Dogs were included in the study if they had had a full clinical examination, at least two thoracic orthogonal radiographic projections (one right lateral and one ventrodorsal/dorsoventral), and an echocardiography performed within 24 h. Two groups were created, a group of control dogs that included dogs older than 1 year of age with no history or concurrent clinical or radiographic signs of cardiovascular or respiratory diseases admitted for a previous study; and a MMVD group that included dogs with an apical systolic heart murmur on auscultation, and a confirmed diagnosed of mitral valve disease consisting of thickening and/or prolapse of the atrioventricular valves and a regurgitant jet on echocardiography. Descriptive data (body weight, age, sex, and breed) were registered. In the control group, dogs with respiratory sinusal arrythmia were included in the study; the presence of other arrythmia or a heart murmur excluded the dog from taking part in the study. Subjects with pulmonary abnormalities, or a history of neck or chest surgery, were not included in the study. Any dog positive to a heartworm antigen test was excluded.

### 2.2. Echocardiography

Every patient had had an echocardiographic exam performed by a veterinary practitioner of our hospital with over 15 years of clinical experience and echocardiography expertise. Examination was carried out without sedation from right and left parasternal position in two-dimensional (2D-) Mode, M-Mode and Doppler Mode, and simultaneous single-lead electrocardiogram, and were performed with a Vivid iq portable ultrasound machine (General Electric Medical Systems, Jiangsu, PR China). From the right parasternal short-axis view, the left ventricle internal diameter at end diastole index to body weight (LVIDDN), measured in accordance with Cornell et al. [45], and left atrium aortic valve ratio (LA/Ao), measured in accordance with Hanson et al. [46], were calculated on an Echopack DICOM viewing system. Dogs with MMVD were classified in different stages based on the echocardiographic criteria recommended by ACVIM guidelines [1]. Dogs in Stage B1 had LVIDDN <1.7 and/or LA/Ao <1.6. Dogs with an increased left atrium, LA:Ao ≥1.6, and left ventricle, LVIDDN ≥1.7, were considered to have LHE. These dogs were included in Stage B2 if asymptomatic, or in Stage C if they had signs of congestive heart failure (tachycardia, respiratory distress, orthopnea, respiratory crackles, and wheezes) current or past.

### 2.3. Radiography

For cardiac measurements, radiographs from the hospital archiving system were retrieved with a digital viewer (IntechForView 12.5.1.1, La Cartuja Baja, Zaragoza, Spain). To be included in the study, the thoracic radiographs had to have been taken in inspiration and with the front limbs extended cranially so the sternal manubrium could be identified. Having vertebral malformations did not exclude the patient from the study. Radiographs with motion artifacts, cardiac silhouette not well defined, or the whole manubrium not included in the image were not included in the study.

Only the right lateral projection was used for the TIHS and VHS measurement. Three measurements, using a digital caliper, were taken and the mean calculated for later analysis.

The VHS was obtained as described by Buchanan et al. [3] and modified according to Jepsen-Grant et al. [47]. The VHS is the sum of the measurements of the long and short cardiac silhouette axes indexed to thoracic vertebral bodies starting at the cranial edge of T4, measured to the nearest 0.1 vertebra (v).

The TIHS was measured following Marbella et al. [8]. First, the lengths of the long axis (LA) and short axis (SA) of the cardiac silhouette were measured, the LA from the central and ventral border of the carina to the cardiac apex, and the SA, perpendicular to the LA, from the cranial border of the cardiac silhouette to the intersection of the ventral border of caudal vena cava with the caudal border of the cardiac silhouette. Second, the thoracic inlet length (TI) was the shortest distance measured from the cranioventral aspect of the first thoracic vertebra to the craniodorsal manubrium. Third, the cardiac axes measurements were added and divided by the TI.

 Radiographic measurements were made by DM. At this time, this investigator was not masked to the clinical data of all the patients and their echocardiographic measurements (Figure 1).

Echocardiographic diagnose and radiographies had to have been performed within the same 24 h.

Intra- and interobserver variability was studied comparing the radiographic measurements of 23 dogs, 15 control and 8 MMVD, randomly selected, performed by two observers (DM and VG). For the intraobserver, variability measurements were taken two times at least one week apart. Observers were blinded to the results of one another. VG did not know what group the patient was included in. The first results of both observers were used to calculate the interobserver coefficient of variation (CV).

### 2.4. Statistical Analysis

The continuous variables providing descriptive information (body weight and age) were presented as median and range (minimum and maximum). For the TIHS and VHS, median, standard deviation, range, and a 95% confidence interval (CI) were calculated on each group. To identify differences in the TIHS depending on sex and body weight (<10 kg, ≥10 kg), differences in the TIHS and VHS between the control dogs versus the dogs with different MMVD stage, and between the different MMVD stages, a paired Student’s *t* test was performed. Also, a Student’s *t* test was performed to compare the TIHS and VHS values of chihuahua dogs included in the different groups studied. Differences with *p* < 0.05 were considered significant. The optimal clinical cutoff value for the radiographic scores was determined based on the highest Youden index ([sensitivity + specificity]−1).

The correlation between the TIHS and VHS was studied with Pearson’s correlation test. It was considered weak when its value was between 0.1–0.3, moderate between 0.4–0.6, strong 0.7–0.9, or perfect 1.

For the intraobserver and interobserver variability an intraclass correlation coefficient >0.9 was considered almost perfect, 0.9–0.7 was considered good, 0.7–0.5 was considered moderate, and <0.5 was considered poor. All statistical analyses were performed using commercially available software (SAS/STAT software, version 16.5, Microsoft Excel 2021).

## 3. Results

A total of 156 dogs were included in the study. The control group had 50 dogs, 28 male and 22 female of different breeds (25 Cross breed, 3 Labrador retriever, 2 each of the breed Golden retriever, Pitbull, and Yorkshire terrier (YT), 1 each of the breed Beagle, Bichon, Border Collie, Bull Terrier, Cavalier King Charles Spaniel (CKCS), Chihuahua, Dachshund, Jack Russell Terrier, Pekingese, Waterdog, Pomeranian, Pug, Schnauzer miniature, Scottish Terrier, Shih-Tzu, Staffordshire, and West Highland White Terrier) with a median age of 4.8 years (range 1–15 years), and mean body weight 7.4 kg (2.10–38.70). Hundred and six dogs of different breeds were diagnosed with MMVD, 58 male and 48 female, with a median age of 11.3 years (5.5–18.9) and mean body weight 5.1 kg (2–26), these being significantly older and lighter than the control dogs, *p* < 0.001 and *p* < 0.01, respectively. Thirty-six dogs were in Stage B1 (15 Cross breed, 5 Chihuahua, 4 YT, 2 each of the breed Cocker Spaniel, Pinscher, and Ratonero, 1 each of the breed French Bulldog, Bull Terrier, Dachshund, Spanish Galgo, Canarian hound, and Pug), 30 in Stage B2 (16 Cross breed, 6 Chihuahua, 4 YT, 2 Spanish Galgo, 1 each of the breed Pekingese, Pinscher), and 40 in Stage C (15 Cross breed, 9 Chihuahua, 6 Bichon, 2 each of the breed CKCS and YT, 1 each of the breed Beagle, Poodle toy, Dalmatian, Pinscher, Shih-Tzu, and Spitz) 23 male, 17 female. There was no difference for age or weight between the different MMVD stages (Table 1).

Normally distributed, the TIHS value for the control group was 2.91 ± 0.23 (2.47–3.44), not different from MMVD Stage B1 2.98 ± 0.36 (2.44–3.65), *p* = 0.12. The TIHS for Stage B2 3.25 ± 0.34 (2.47–4.18) and Stage C 3.53 ± 0.36 (3.20–4.83) increased compared to the control group and Stage B1, *p* < 0.0001 (Table 2). There was a statistically significant difference between Stage B1 and Stage B2 and C, *p* = 0.002 and *p* < 0.0001, respectively. The TIHS was significantly different between Stage B2 and Stage C *p* = 0.002 (Figure 2). There was no difference for the TIHS between the sex or body weight in any group, *p* > 0.05 (Table 3).

Chihuahua dogs, 32 patients, were the most represented breed in our hospital. Twelve were in the control group and 20 in MMVD group: 5 Stage B1, 6 Stage B2, and 9 Stage C. There was no difference for the TIHS nor for the VHS between the chihuahua dogs and the rest of the population neither on the control group nor on either of the MMVD groups *p* > 0.05. The TIHS for the chihuahua breed did not show differences between the control group 2.96 ± 0.33 (2.33–3.60) and Stage B1 2.92 ± 0.11 (2.81–3.06). The TIHS value increased with MMVD stage, B2 3.40 ± 0.28 (2.99–3.83) and C 3.61 ± 0.44 (2.97–4.37). The difference was statistically significant between the control group and Stage B2 (*p* = 0.015) and C (*p* = 0.0008). There was also a difference statistically significant between Stage B1 and B2 and C, *p* = 0.0052 and *p* = 0.0034, respectively (Table 4).

For the entire studied population, a TIHS value of 3.3 had a sensitivity of 69% and specificity of 88% to discriminate between dogs with and without cardiac enlargement, Youden Index 0.57 (Table 5). The same value had a sensitivity of 69% and specificity of 81% to detect cardiac remodeling (stage B2 and C) in dogs with MMVD, YI 0.50. At the same time, a VHS value of 11.5 v showed a sensitivity of 37% and specificity of 97% to discriminate between MMVD dogs with and without cardiac remodeling, YI 0.34 (Table 6) The area under the curve for the THIS and VHS was good, 0.82 and 0.83, respectively (Figure 3). There was a moderate correlation between the TIHS and VHS, 0.59 (Figure 3).

The intraobserver variability was almost perfect for the TIHS in the 23 dogs compared, also considering only the 15 control dogs and the 8 MMVD dogs, 0.97, 0.93, and 0.96, respectively. The same accounts for VHS of 0.98, 0.99, and 0.96. Interobserver variability was good for TIHS of 0.78 and VHS of 0.87 for the 23 dogs compared, 0.77 and 0.84, considering only the 15 control dogs. And it was good for the VHS, 0.84, but poor for TIHS of 0.43, in the MMVD (Figure 4).

## 4. Discussion

The TIHS method can be used to identify dogs with cardiac enlargement secondary to MMVD. A TIHS cutoff of 3.3 detected 69% of the dogs with cardiac enlargement and 81% of the dogs that did not have cardiac enlargement. A previous study showed that 90% of clinically healthy dogs had a TIHS < 3.2 [8]. In this study, 86% (43/50) of normal dogs had a TIHS < 3.2, and that percentage decreased in the dogs with MMVD as the disease worsened, 66% (24/36) Stage B1, 46% (14/30) B2, and 10% (4/40) Stage C.

For the overall population, the control group was heavier than any MMVD group and its TI was longer compared to Stage C dogs, *p* = 0.003. Only 4 dogs in Stage C weighted more than 10 kg compared to 23 dogs in control group. The TI, SA, LA length, and SA + LA were longer for dogs ≥10 kg compared to dogs <10 kg independently of the group, the heavier the dog the longer the heart axes and thoracic inlet length. Comparing different groups, SA and LA and their sum were higher for dogs in Stage B2 and C versus normal dogs, independently of the weight. On the contrary, there was no difference for the TI between the different groups depending on the weight. In dogs <10 kg, the TIHS increased significantly from the control group to Stage C, in every group respect to the previous; control vs. Stage B1 (*p* < 0.04), B2 (*p* < 0.0001) and C (*p* < 0.0001), Stage B1 vs. B2 (*p* < 0.02) and C (*p* < 0.0001), and Stage B2 vs. C (*p* < 0.007). The difference in the TIHS was statistically significantly for dogs ≥10 kg between the control group and Stage B2 (*p* < 0.01) and C (*p* < 0.0001), and Stage B1 and C (*p* < 0.005). According to our results, the difference observed in the TIHS value between groups was due to the increase in heart short and long axes length secondary to mitral valve disease as the disease progresses.

The TIHS value increased with the disease stage as did the VHS. This could be expected as the cardiac axes measurement in both methods are the same. The increase in the VHS observed in this study is in accordance with previous studies in dogs with MMVD [2,20] and it is associated with the stage of the disease [2]. One study assessed the increase in the VHS before the onset of congestive heart failure [48]. In the present study, the TIHS increased with disease stage. It seems that the TIHS could be used to monitor the disease progression in a patient, but a longitudinal study would be ideal to confirm this point.

A VHS ≥ 11.5 v, a value that could be considered evidence of cardiomegaly [1], showed lower sensitivity 0.37 and YI 0.31 compared to a TIHS ≥ 3.3, Se 0.69, and YI 0.50, though its specificity was higher, 0.94 vs. 0.81. Previous studies evaluating radiographic predictors for detection of left heart enlargement in dogs with MMVD have proposed different VHS cutoffs [16,17,18,20]. In their study, Levicar et al. found that a VHS > 11 v had an 82% sensitivity and 71% specificity, AUC 0.82, to discriminate between B1 and B2 dogs [20]. Duler et al. indicated in their research that a VHS > 11.1 v had a 65.5% sensitivity and 80% specificity, AUC 0.78, to detect Stage B2 [18]. In our population, a VHS >11 v showed a lower sensitivity 40% but similar specificity 75%, AUC 0.74, to discriminate between Stage B1 and B2 dogs. The maximum specificity was a VHS >12 v, 97%. Other studies have found similar specificity for a cutoff >12 v [17,18]. The differences observed between different studies might be related to breed variability within the studied populations.

In the present study, a VHS ≥11.5 v would not identify cardiac enlargement in 63% (44/70) compared to 31% (22/70) for a TIHS ≥ 3.3, of the dogs diagnosed with cardiac enlargement on echocardiography. A lower VHS, ≥11 v, showed higher sensitivity 0.73, than a TIHS ≥ 3.3, 0.68. It did not identify cardiac enlargement in fewer dogs that did have it compared to a TIHS ≥3.3, 29% (19/70) and 31% (22/70), respectively. However, it diagnosed left heart enlargement in more dogs that did not have it than a TIHS ≥3.3, 33% (12/36) compared to 19% (7/36). A high specificity, meaning few false positive, is important because discriminating between B1 and B2 is associated with the prescription of lifelong treatment. On the other hand, a low sensitivity would deny treatment to some dogs that might benefit from it.

Although the population size was small, we evaluated the TIHS in Chihuahuas. Dogs of this breed are frequently diagnosed with MMVD [2,36,49]. A cutoff TIHS ≥3.3 discriminated between chihuahua dogs in Stage B1 and B2 with higher specificity than a VHS ≥11.5, 83% and 62%, respectively. Whether there are breed differences in the TIHS value needs further study.

The correlation between the TIHS and VHS was moderate 0.60. Correlation between the TIHS and echocardiographic measurements, LVIDDN and LA/Ao, was also moderate, 0.62 and 0.42, respectively. Correlation between the VHS and LVIDDN and LA/Ao was moderate, 0.56 and 0.47, respectively. These results vary compared to other studies, being higher to the results of Stepien et al.’s study [17] but lower than the results by Lam et al. [13] and Duler et al. [18]. Not surprisingly, correlation was higher for the left ventricle. As shown on angiocardiographic views, the heart axes combined include the right and heart chambers [3]. CT images have shown that this is especially true for the short axis, but the left atrium is not specifically included [50]. More specific methods to assesses the left atrium radiographically like VLAS, M-VLAS, RLAD have shown higher positive correlation with echocardiographic values than the VHS [11,13,18,23], though others have not [39]. Comparison of those methods with the TIHS was not a goal of this research, and it would need further study.

Interobserver agreement for the 23 dogs compared was better for the VHS than the TIHS, 0.87 and 0.78, respectively. The interobserver agreement in the VHS for the MMVD dogs, 0.85, was lower compared to previous studies with MMVD dogs, ranging between 0.96 [51] and 0.92 [52]. The agreement between observers in the TIHS for the control group was 0.77. Interestingly, the interobserver variability showed poor agreement for the MMVD dogs when compared, 0.43. However, there was no statistically significant difference in the TI, LA, and SA measurements between observers. The identification of reference points in dogs with MMVD could be hampered by the presence of congestive heart failure (6/7 dogs with MMVD compared were in Stage C). The TIHS method being a ratio can change with a small difference in some of the magnitudes that take part in its calculation when measured by different observers. Previous studies have concluded that the VHS was independent of the observer’s experience [51,52] but dependent of individual observer. This could be the case for the TIHS too, as it is an adaptation of the VHS method. As with the VHS, the ideal clinical scenario would be that the same practitioner did the follow up in a particular case [52].

This study has some limitations. Although echocardiographic exams were not performed by the same operator, standardized methods were used to obtain echocardiographic measurements and staging of dogs with MMVD. Thus, the heart chambers’ measurements could be subjected to interobserver variability. The main observer was not blinded to the dog’s clinical status and echocardiographic results, which could have caused radiographic measurement bias. The recommendation for the treatment of MMVD dogs included in this study followed ACVIM guidelines, adapted for every case. Thus, diuretic dosages could be different between dogs for the diuretic drug used (furosemide or torasemide), or the administration of spironolactone or amlodipine. Treatment changes the VHS in the first 6 months of treatment [53]. The TIHS being a variant of the VHS could be affected in a similar way, and this needs further study. Our study population of the control dogs and the dogs with MMVD were not homogenous and included different a number of dogs of different breeds, reflecting a general population of dogs attending a veterinary hospital. The effect of breed variability on the echocardiographic [54,55] and radiographic measurements has been studied [24,25,26,27,28,29,30,31,32,33,34,35,36,37,38,39,40]. This needs to be considered when conducting radiographic follow up of dogs with MMVD. Dog breeds can have different thorax conformation, and how this affects the TIHS needs further study. Radiographs were intended to be taken during peak inspiration, but this is not always possible in awake animals. Caudal thorax diameter varies with the respiratory cycle, and whether respiration changes the thoracic inlet length has not been studied, nor how this could affect the TIHS value. In this population, this method has shown a low interobserver agreement for MMVD dogs. Further studies with larger populations and more observers are desirable.

## 5. Conclusions

The TIHS is a simple method to measure the cardiac silhouette in dogs with and without myxomatous mitral valve disease. A TIHS value >3.3 would suggest cardiac enlargement in a dog with a heart murmur secondary to mitral valve disease, recommending an echocardiography. The TIHS can help clinicians in the staging of MMVD in a general population of dogs, including dogs with midthoracic vertebral anomalies, or when the thoracic vertebral bodies cannot be identified, and of chihuahua dogs, a breed commonly affected by the disease. It could also be used to monitor progression of the disease in a dog already diagnosed.

## Figures and Tables

**Figure 1 animals-13-02666-f001:**
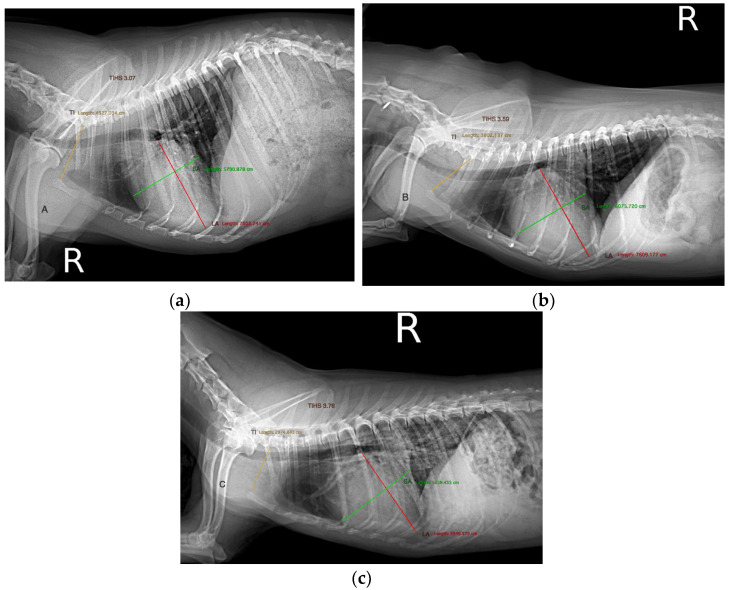
Thoracic inlet heart score measure in three dogs diagnosed with myxomatous mitral valve disease in different stages. (**a**) Stage B1, (**b**) Stage B2, (**c**) Stage C. LA heart long axis, SA heart short axis, TI thoracic inlet length. THIS = (SA + LA)/TI.

**Figure 2 animals-13-02666-f002:**
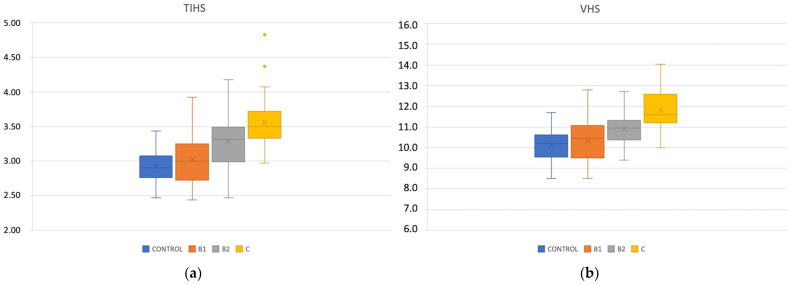
Box plots illustrating TIHS (**a**) and VHS (**b**) for dogs included in the study according to their clinical status. MMVD stages B1, B2, and C.

**Figure 3 animals-13-02666-f003:**
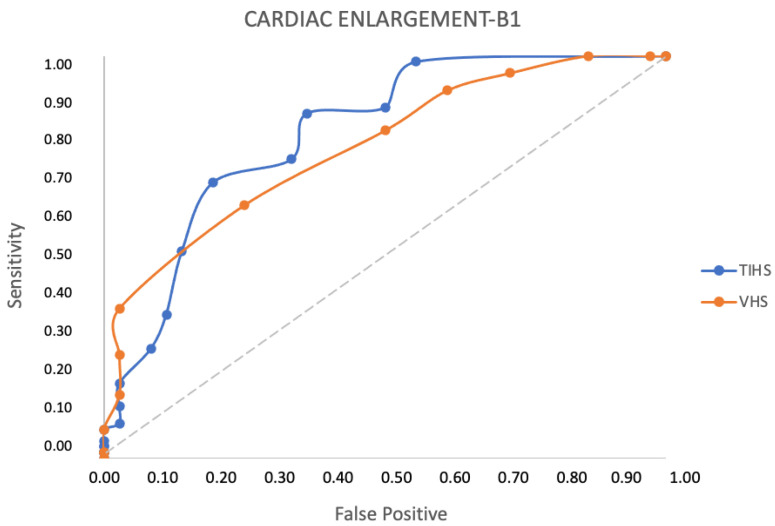
Comparison of the receiver operating characteristics curves for thoracic inlet heart score (TIHS) and vertebral heart score (VHS) for detecting cardiac enlargement in 106 dogs with myxomatous mitral valve disease.

**Figure 4 animals-13-02666-f004:**
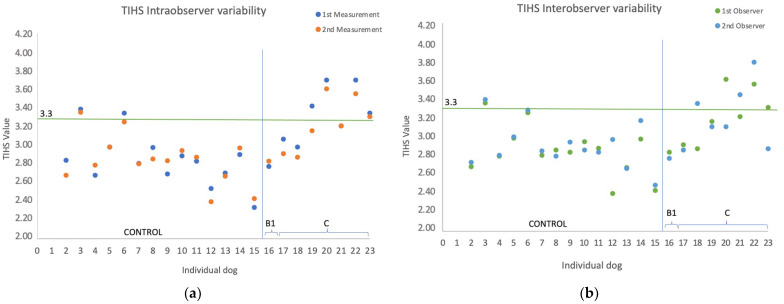
Scatterplots for intra- and interobserver variability on TIHS (**a**,**b**) and VHS (**c**,**d**) values measured on right lateral radiographic projections from 23 randomly selected dogs, 15 control and 8 MMVD. The horizontal line represents the cutoff value. The vertical line separates control dogs from MMVD dogs.

**Table 1 animals-13-02666-t001:** Mean and range values of descriptive variables (age and weight) of the studied dogs in the different groups. Mean and standard deviation for radiographic and echocardiographic measurements.

n	Control 50	Stage B1 36	Stage B2 30	Stage C 40
Age (years)	4.8 (1–15.3) ^a^	11.6 (5.5–15.9)	11.3 (7.4–18.9)	11.4 (7.9–17.6)
Weight (kg)	7.40 ^b^ (2.10–38.70)	5.38 (2.60–26.0)	5.18 (2.0–19.60)	4.73 (2.80–14.80)
Male/Female	28/22	18/18	17/13	23/17
TI	29.15 ± 15.11 ^c^	25.91 ± 9.50	26.25 ± 9.55	24.13 ± 7.39
SAx	39.24 ± 20.60	36.51 ± 12.53	39.84 ± 13.73	40.44 ± 11.03
LAx	45.38 ± 26.81	42.09 ± 16.96	46.23 ± 17.64	45.64 ± 12.13
SAx + LAx	87.68 ± 50.97	78.75 ± 29.30	86.24 ± 31.16	86.25 ± 22.84
LA/Ao	1.20 ± 0.18	1.36 ± 0.17	1.80 ± 0.22 ^d^	2.08 ± 0.36 ^d^
LVIDDN	1.57 ± 0.20	1.57 ± 0.21	1.94 ± 0.24 ^d^	2.08 ± 0.29 ^d^

^a^ Difference statistically significant between control and MMVD groups. ^b^ Difference statistically significant for weight between control and MMVD groups. ^c^ Difference statistically significant for TI between control and MMVD Stage C. ^d^ Difference statistically significant for left atrium and left ventricle compared to the previous group. *p* < 0.05. LA/Ao ratio left atrium aorta. LVIDDN left ventricular internal diameter in diastole normalized to body weight, LA long axis, n number of dogs, SA short axis, TI thoracic Inlet.

**Table 2 animals-13-02666-t002:** Radiographic cardiac size measurements in the control dogs and dogs diagnosed with myxomatous mitral valve disease (MMVD) in different stages.

n	Control 50	Stage B1 36	Stage B2 30	Stage C 40	*p*
TIHS (CI)	2.91 ± 0.23 (2.84–2.98)	2.98 ± 0.36 ^a^ (2.87–3.11)	3.25 ± 0.34 ^a^ (3.09–3.39)	3.53 ± 0.36 ^a^ (3.42–3.64)	<0.01
VHS (CI)	10.07 ± 0.73 (9.86–10.28)	10.24 ± 0.95 ^b^ (9.93–10.55)	10.83 ± 0.86 ^b ^ (10.49–11.25)	11.74 ± 0.96 ^b^ (11.43–12.05)	0.016

^a^ Difference statistically significant for TIHS between control group and MMVD Stage B2 and C, and MMVD groups compared with the previous stage (Stage B1 and Stage B2 and C, and Stage B2 and C). ^b^ Difference statistically significant for VHS between control group and MMVD Stage B2 and C, and MMVD groups compared with the previous stage (Stage B1 and Stage B2 and C, and Stage B2 and C). *p* < 0.05. CI confidence interval 95%, n number of dogs, TIHS thoracic inlet heart score. VHS vertebral heart score.

**Table 3 animals-13-02666-t003:** TIHS value according to sex and body weight in the different groups. There was no difference statistically significant in any group, *p* < 0.05. n number of dogs.

THIS (n)	Male	Female	*p*	<10 kg	≥10 kg	*p*
CONTROL	2.88 ± 0.23 (28)	2.94 ± 0.23 (22)	0.33	2.88 ± 0.24 (27)	2.94 ± 0.22 (23)	0.41
Stage B1	3.01 ± 0.35 (18)	2.95 ± 0.38 (18)	0.62	3.04 ± 0.35 (26)	2.84 ± 0.0.35 (10)	0.13
Stage B2	3.25 ± 0.38 (17)	3.26 ± 0.29 (13)	0.91	3.26 ± 0.37 (25)	3.22 ± 0.18 (5)	0.51
Stage C	3.53 ± 0.37 (23)	3.53 ± 0.35 (17)	0.99	3.53 ± 0.37 (36)	3.50 ± 0.21 (4)	0.78

**Table 4 animals-13-02666-t004:** Echocardiographic measurements of the left atrium and the left ventricle internal diameter in diastole, and radiographic cardiac size measurements in a population of chihuahua dogs.

Chihuahua n	Control 12	Stage B1 5	Stage B2 6	Stage C 9	*p*
LA/Ao	1.31 ± 0.01 ^a^	1.33 ± 0.15 ^a^	1.86 ± 0.18	2.07 ± 0.52	≤0.001
LVIDDN	1.37 ± 0.06 ^b^	1.50 ± 0.24 ^b^	1.87 ± 0.31	2.12 ± 0.28	≤0.001
TIHS (CI)	2.96 ± 0.33 ^c^ (2.77–3.15)	2.92 ± 0.11 ^c^ (2.87–3.11)	3.40 ± 0.28 (3.10–3.70)	3.61 ± 0.44 (3.27–3.95)	≤0.01
VHS (CI)	9.82 ± 0.60 ^d^ (9.44–10.2)	10.33 ± 0.65 ^d^ (9.76–10.90)	11.26 ± 0.71 (10.52–12.0)	12.08 ± 1.06 (11.27–12.89)	≤0.001

^a^ Difference statistically significant between dogs with (B2 and C) and without (control and B1) cardiac enlargement. ^b^ Difference statistically significant between dogs without cardiac enlargement and Stage C. ^c^ Difference statistically significant for TIHS between dogs with and without cardiac enlargement. ^d^ Difference statistically significant for VHS between control group and Stage B2 and Stage C, and Stage B1 compared to Stage C. *p* < 0.05. LA/Ao ratio left atrium aorta, LVIDDN left ventricular internal diameter in diastole normalized to body weight, n number of dogs, TIHS thoracic inlet heart score, VHS vertebral heart score.

**Table 5 animals-13-02666-t005:** Statistical values for different TIHS and VHS cutoff to differentiate between B1 and B2 dogs. AUC area under the curve. NPV negative predictive value, PPV positive predictive value, Se sensitivity, Sp specificity.

Radiographic Method	AUC (95% CI)	Se	Sp	Youden Index	PPV	NPV	Cutoff
TIHS	0.75	0.53 0.53 0.43	0.75 0.81 0.81	0.28 0.33 0.24	0.64 0.70 0.65	0.66 0.67 0.63	≥3.25 ≥3.30 ≥3.35
VHS (v)	0.74	0.50 0.20 0.13	0.66 0.94 0.97	0.16 0.14 0.10	0.65 0.75 0.80	0.63 0.59 0.57	≥11.0 ≥11.5 ≥12.0

**Table 6 animals-13-02666-t006:** Diagnostic accuracy for different TIHS and VHS cutoff values to differentiate between MMVD dogs with (B2 and C) and without cardiac enlargement (B1). AUC area under the curve. NPV negative predictive value, PPV positive predictive value, Se sensitivity, Sp specificity.

Radiographic Method	AUC (95% CI)	Se	Sp	Youden Index	PPV	NPV	Cutoff
TIHS	0.82	0.70 0.69 0.60	0.75 0.81 0.81	0.45 0.50 0.41	0.84 0.87 0.86	0.56 0.57 0.51	≥3.25 ≥3.30 ≥3.35
VHS (v)	0.83	0.73 0.37 0.21	0.66 0.94 0.97	0.40 0.31 0.26	0.81 0.93 0.95	0.56 0.44 0.41	≥11.0 ≥11.5 ≥12.0

## Data Availability

The data presented in this study are available on request from the corresponding author.

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
