# Peer review of "The Thoracic Inlet Length as a Reference Point to Radiographically Assess Cardiac Enlargement in Dogs with Myxomatous Mitral Valve Disease"

_animals, 2023, doi:10.3390/ani13162666_

Round 1
Reviewer 1 Report
I found the topic of the study to be relevant and the manuscript to be interesting. Still, I think that the clarity of the manuscript should be improved in some points. The radiographic images in the manuscript were too small for me to evaluate and were not included in any other form in the review.
Lines 42-44: As radiography is actually part of the staging as proposed in the ACVIM consensus statement, I would consider changing the text as follows: “It is also used for staging the disease in conjunction with auscultation and radiography [1]. Radiography can also assist…”
Line 48: should read “TIHS” instead of “THIS”
Line 101 Echocardiography
Echocardiographic Criteria for staging of MVD are given. Please also include criteria for the diagnosis of the disease (e. g. regurgitation, mitral valve thickening…).
Please specify if results of auscultation and thoracic radiography were also considered in disease staging, as recommended in the ACVIM consensus (VHS > 10,5 or breed specific, heart murmur ≥ 3/6), or if staging was only based on echocardiography as part of the study design.
Please clarify which criteria were used as the gold standard to determine, whether a dog had cardiomegaly. I assume it’s (only) the echocardiographic criteria (LA/Ao ≥ 1,6 and LVIDDN ≥ 1,7)?
I did not understand the reason for the comparison between Chihuahuas and dogs of other breeds (Lines 219 - 228). This comparison did not seem to promote the goals of the study in any way. I would have been more interested in information regarding the prevalence of vertebral malformations and whether their presence influences the correlation between VHS and TIHS in affected dogs.
I don’t understand Figure 3. Please clarify the content and the relevance of the figure.
Limitations:
Differences in patient size between control and disease group (lines 181-182, table 1) had to be expected, but they should be included in the limitations section, as differences between MVD- and control-group could influence the results.
Author Response
Thank you very much for your comments and for helping us to improve this manuscript.
Firstly, I am sorry to hear that you could not evaluate the radiographic images properly. In our discharge, they were provided as separate files.
I will try to address each one of your comments separately.
Lines 42-44: As radiography is actually part of the staging as proposed in the ACVIM consensus statement, I would consider changing the text as follows: “It is also used for staging the disease in conjunction with auscultation and radiography [1]. Radiography can also assist…”
Thank you for the suggestion. Added to the text, lines 42-43.
Line 48: should read “TIHS” instead of “THIS”.
Changed, line 49.
Line 101 Echocardiography
Echocardiographic Criteria for staging of MVD are given. Please also include criteria for the diagnosis of the disease (e. g. regurgitation, mitral valve thickening…).
The criteria for the diagnosis of MVD are included on lines 94-97.
Please specify if results of auscultation and thoracic radiography were also considered in disease staging, as recommended in the ACVIM consensus (VHS > 10,5 or breed specific, heart murmur ≥ 3/6), or if staging was only based on echocardiography as part of the study design.
Only echocardiographic criteria were considered for the staging. Different studies have proposed different VHS to discriminate between stage B1 and B2. Although, the ACVIM guidelines recommend a VHS >10.5 or breed specific alongside a heart murmur ≥ 3/6 and the echocardiographic values, it also states that in the absence of echocardiography a VHS ≥ 11.5 would identify dogs with cardiac enlargement. So far, echocardiographic criteria are the only ones that seem to be “precise or more reliable”, although recent studies suggest that what is considered left ventricle enlargement, LVIDDN≥1.7, or left atrial enlargement, LA/Ao ≥1.6, could be normal for some breeds. Rishniw. M.; Brown, D. The ACVIM consensus statement definition of left ventricular enlargement in myxomatous mitral valve disease does not always represent left ventricular enlargement. J. Vet. Cardiol. 2022 Aug, 42, pp. 92-102.
Rishniw, M.; Caivano, D.; Dickson, D.; Vatne, L.; Harris, J.; Matos, J.N. Two-dimensional echocardiographic left- atrial-to-aortic 498 ratio in healthy adult dogs: a reexamination of reference intervals. J. Vet. Cardiol. 2019, 26, pp. 29-38.
Please clarify which criteria were used as the gold standard to determine, whether a dog had cardiomegaly. I assume it’s (only) the echocardiographic criteria (LA/Ao ≥ 1,6 and LVIDDN ≥ 1,7)?
Yes, a dog was considered to have left heart enlargement based on the echocardiographic findings, LA/Ao ≥ 1,6 and LVIDDN ≥ 1,7
I did not understand the reason for the comparison between Chihuahuas and dogs of other breeds (Lines 219 - 228). This comparison did not seem to promote the goals of the study in any way. I would have been more interested in information regarding the prevalence of vertebral malformations and whether their presence influences the correlation between VHS and TIHS in affected dogs.
The idea behind the chihuahua dog comparison was to evaluate if THIS is affected by breed, in the control group and in dogs with MVD. Our results show that TIHS increases as the disease progress in the general population and in one specific breed, chihuahuas. And the TIHS value does not show differences between the general population and chihuahuas in any of the groups. So, it could be used to assess the progression of the disease of any dog independently of the breed. Having said that, the chihuahua population was small and a larger sample would have been ideal. More breeds with larger samples studies are needed to assess whether TIHS is affected by breed. This is addressed in line 331-332.
Regarding vertebral malformations, some studies have shown how VHS is affected by its presence, usually increasing it. As you say, it would have been interesting comparing how that could affect the correlation between VHS and THIS. However, we did not find many dogs with vertebral anomalies so we could make conclusion about that aspect.
I don’t understand Figure 3. Please clarify the content and the relevance of the figure.
This is a Receiver Operating Curve to assess the accuracy of a method, in this case two radiographic methods to identify cardiac enlargement on radiographies, based on the Area Under the Curve. It relevance is explained on the text 251-253.
Limitations:
Differences in patient size between control and disease group (lines 181-182, table 1) had to be expected, but they should be included in the limitations section, as differences between MVD- and control-group could influence the results.
Addressed lines 369-371.
Reviewer 2 Report
The article deals with an interesting and important, practical topic. X-ray detection of MMVD is widely used and facilitates and reduces the cost of diagnosis. The limitation of this study was certainly the large number and therefore small populations of individual breeds. In the control group and ACVIM B1 there were dogs of both large and small breeds.
Since the age of the animals was varied, it would be worth writing on what basis patients of large breeds were classified as MMVD patients. We know that this disease can occur in large breeds and often they do not show typical valve changes. If these were asymptomatic dogs aged 2-3 years, in which mitral regurgitation was detected only in echocardiography, we are not sure whether they were dogs with MMVD and not, for example, with congenital slight dysplasia of this valve.
It is also worth adding in the limitations of the study in the discussion that no dogs had different chest shapes, both barrel-chested and deep chested dogs, and we know that this affects the shape of the heart and thus the VHS measurements.
Author Response
The article deals with an interesting and important, practical topic. X-ray detection of MMVD is widely used and facilitates and reduces the cost of diagnosis. The limitation of this study was certainly the large number and therefore small populations of individual breeds. In the control group and ACVIM B1 there were dogs of both large and small breeds.
Thank you for your comments and your help to improve this manuscript.
As you mention, the small population of individual breeds is a limitation of the study, but the large number of dogs adds value to the results and reflects a general population attending a veterinary hospital any given day.
Since the age of the animals was varied, it would be worth writing on what basis patients of large breeds were classified as MMVD patients. We know that this disease can occur in large breeds and often they do not show typical valve changes. If these were asymptomatic dogs aged 2-3 years, in which mitral regurgitation was detected only in echocardiography, we are not sure whether they were dogs with MMVD and not, for example, with congenital slight dysplasia of this valve.
The youngest dog diagnosed with MVD was a cross-breed dog 66 months old (five and a have years), 4.7 kg heavy. She showed on echocardiography the features of mitral valve disease, thickening of the valve and regurgitation. As you say, this could still be mitral valve dysplasia, however, this was a patient of the hospital since she was a puppy, and no heart murmur had been auscultated previously during her annual booster examination.
It is also worth adding in the limitations of the study in the discussion that no dogs had different chest shapes, both barrel-chested and deep chested dogs, and we know that this affects the shape of the heart and thus the VHS measurements.
I agree with your comment, a line has been added in limitations. Line 374-375.
VHS can be different in dogs with different chest shapes, but in dogs considered to be deep-chested, Doberman 10.0v (Lamb et al, 2001), Whippet 11.3v (Bavegens et al, 2005), Greyhound 10.5v (Marin et al, 2007), the VHS have shown to be different. From my point of view, VHS is more affected by the breed itself than by the shape of the thorax, and it could be related to the length of the vertebral bodies and the intervertebral space. Do breeds related, shepherd dogs, hound dogs, retrievers, terriers… have the same or different VHS, THIS? This could be an interesting study.
Reviewer 3 Report
This is a very interesting work dealing with the TIHS index in normal dogs and in dogs with MMD. Despite the fact that the sensitivity and specificity is not high enough, it seems that TIHS index could be a useful index descriminating the normal and B1 stage MVD dogs from dogs with B2 and C stages. I will suggest to authors to continue working on this area in order to improve the quality of this index.
I
Author Response
This is a very interesting work dealing with the TIHS index in normal dogs and in dogs with MMD. Despite the fact that the sensitivity and specificity is not high enough, it seems that TIHS index could be a useful index descriminating the normal and B1 stage MVD dogs from dogs with B2 and C stages. I will suggest to authors to continue working on this area in order to improve the quality of this index.
Thank you for your comments and for the encouragement.